# Effect of Short-Term Dietary Intervention and Probiotic Mix Supplementation on the Gut Microbiota of Elderly Obese Women

**DOI:** 10.3390/nu11123011

**Published:** 2019-12-10

**Authors:** Raffaella Cancello, Silvia Turroni, Simone Rampelli, Stefania Cattaldo, Marco Candela, Laila Cattani, Stefania Mai, Roberta Vietti, Massimo Scacchi, Patrizia Brigidi, Cecilia Invitti

**Affiliations:** 1Department of Medical Sciences and Rehabilitation, IRCCS Istituto Auxologico Italiano, 20145 Milan, Italy; 2Department of Pharmacy and Biotechnology, Unit of Microbial Ecology of Health, University of Bologna, 40126 Bologna, Italy; 3Laboratory of Clinical Neurobiology, IRCCS Istituto Auxologico Italiano, 28824 Piancavallo (VB), Italy; 4Division of Nutritional Rehabilitation, IRCCS Istituto Auxologico Italiano, 28824 Piancavallo (VB), Italy; 5Laboratory of Metabolic Research, IRCCS Istituto Auxologico Italiano, 28824 Piancavallo (VB), Italy; 6Division of Endocrinology and Metabolic Diseases, IRCCS Istituto Auxologico Italiano, 28824 Piancavallo (VB), Italy; 7Department of Clinical Science and Community Health, University of Milan, 20100 Milan, Italy

**Keywords:** gut microbiota, elderly, obesity, Mediterranean diet, probiotics

## Abstract

Accumulating literature is providing evidence that the gut microbiota is involved in metabolic disorders, but the question of how to effectively modulate it to restore homeostasis, especially in the elderly, is still under debate. In this study, we profiled the intestinal microbiota of 20 elderly obese women (EO) at the baseline (T0), after 15 days of hypocaloric Mediterranean diet administered as part of a nutritional-metabolic rehabilitation program for obesity (T1), and after a further 15 days of the same diet supplemented with a probiotic mix (T2). Fecal samples were characterized by Illumina MiSeq sequencing of the 16S rRNA gene. The EO microbiota showed the typical alterations found in obesity, namely, an increase in potential pro-inflammatory components (i.e., *Collinsella*) and a decrease in health-promoting, short-chain fatty acid producers (i.e., *Lachnospiraceae* and *Ruminococcaceae* members), with a tendency to reduced biodiversity. After 15 days of the rehabilitation program, weight decreased by (2.7 ± 1.5)% and the gut microbiota dysbiosis was partially reversed, with a decline of *Collinsella* and an increase in leanness-related taxa. During the next 15 days of diet and probiotics, weight dropped further by (1.2 ± 1.1)%, markers of oxidative stress improved, and *Akkermansia*, a mucin degrader with beneficial effects on host metabolism, increased significantly. These findings support the relevant role of a correct dietetic approach, even in the short term, to modulate the EO gut microbiota towards a metabolic health-related configuration, counteracting the increased risk of morbidity in these patients.

## 1. Introduction

Individuals aged 60 years and older are expected to nearly double from 12% to 22% of the global population between 2015 and 2050 [1]. Furthermore, obesity has increased at any age, and old obese individuals have nearly tripled between 1975 and 2016 [2], potentially increasing geriatric syndromes related to the coexistence of obesity with advanced age. Identifying effective strategies that guarantee the best health status of elderly people is imperative to improve public health and reduce societal costs.

Elderly subjects present a physiological reduction of several biological functions, one of the most common being the impairment of the gastrointestinal tract function. Masticatory function, taste and saliva production impair with age, and gastric acid secretion and gastrointestinal transit time slow down, leading to constipation [3]. The impairment of these biological functions is held responsible for the frequent adoption of an unbalanced diet with impaired nutritional status, and, together with this, believed to contribute to the typical age-related changes in the gut microbiota [4]. The intestinal microbial ecosystem of older people is in fact commonly characterized by a reduction in biodiversity (an accepted hallmark of a healthy gut), enrichment in opportunistic pathogens (e.g., enterobacteria) and a decrease in saccharolytic bacteria from the *Lachnospiraceae* and *Ruminococcaceae* families, capable of producing short-chain fatty acids (SCFAs), particularly butyrate [5]. Given the multifaceted role of SCFAs in human physiology (e.g., maintenance of epithelial barrier integrity, metabolic regulation, immune functioning and modulation of neuronal activity) [6], this aged-type microbiota profile is supposed to variously contribute to the age-related functional decline, supporting inflammaging (i.e., the low-grade chronic inflammation characterizing the advancement of age) [7] and thus contributing to the pathophysiology of inflammation-related diseases frequently observed in the elderly [8,9,10,11].

An additional factor that may potentially exacerbate the aging-associated inflammation is the accumulation of visceral adipose tissue with advancing age [12,13]. It is well established that obesity is associated with dysbiotic microbial layouts, featuring reduced species richness and increases in the proportions of pathobionts and sulphate-reducing bacteria [14,15,16]. These microbiome signatures are recognized to contribute to (and possibly predict) the obese phenotype in a multifactorial way, i.e., supplying extra calories to the host, affecting satiety, favoring fat storage, disrupting the integrity of the epithelial barrier and consolidating the inflammatory state [17,18,19]. In light of the role of diet as a pivotal determinant of gut microbiota structure and function [20], targeted microbiota manipulation via nutritional interventions has recently been promoted as a novel ecological approach for the prevention and treatment of obesity and related metabolic disorders, as well as potentially of other dysbiosis-related diseases [17,21]. However, despite encouraging results [22,23,24], several questions still remain unanswered, especially if and to what extent a dietary intervention can reverse the gut microbiota dysbiosis in morbidly obese elderly subjects, with ultimate positive impact on health outcomes. No less important, studies of this type are usually complicated by many confounding variables, such as the frequent use by aged individuals of drugs potentially affecting the microbiota, physical activity and general exposome-related factors.

In an attempt to bridge this gap, here we analyzed the gut microbiota composition in a group of elderly obese women, without bowel disease and not taking well-known microbiota-modifying drugs, before, during and after one month of a metabolic-nutritional-psychological rehabilitation program for obesity. During this period, hospitalized patients received a hypocaloric Mediterranean diet administered alone for two weeks and enriched with a probiotic mix in the next two weeks, and they followed an individual physical activity program. Biochemical parameters, including metabolic, inflammatory and oxidative stress markers, were measured throughout the intervention as well. According to our findings, 15 days of dietary intervention in older people with moderate to severe obesity, as part of a metabolic-nutritional-psychological rehabilitation program, confer metabolic improvements and impact on microbiota dysbiosis towards a profile typically associated with metabolic health.

## 2. Materials and Methods

### 2.1. Study Cohort, Metabolic-Nutritional-Psychological Rehabilitation Program, and Data and Sample Collection

Twenty elderly obese (EO) women aged ≥65 years were recruited between 2016 and 2018 from obese patients admitted to the IRCCS, Istituto Auxologico Italiano (Piancavallo-Verbania, Italy), for a 1-month metabolic-nutritional-psychological rehabilitation program. Exclusion criteria were: (1) history of colorectal surgery or bariatric surgery; (2) known or suspected inflammatory bowel disease and/or proctitis; (3) prebiotic, probiotic and/or vitamin supplementation in the previous three weeks; (4) therapy with antibiotics, proton pump inhibitors and/or metformin; (5) previous or concomitant history of cancer; (6) inability to perform physical activity.

The rehabilitation program consisted of individual sessions for nutritional education, peer group psychological support and physical activity. Patients received a balanced hypocaloric Mediterranean diet for the first 15 days, enriched with a probiotic mix in the following 15 days. The administered Mediterranean diet was based on fresh foods low in salt (≤3 g/day) and simple sugars, able to induce an average daily energy deficit of 250 kcal. Diet composition was: 18–20% protein, 27–30% fat (<8% saturated fat) and 50–55% carbohydrate (<15% simple sugars), and 30 g of fibers from vegetables. The VSL#3 probiotic mixture (Ferring Pharmaceuticals, West Drayton, UK) was chosen because it contains eight different strains of lactic acid-producing bacteria, i.e., *Streptococcus thermophilus*, bifidobacteria (*Bifidobacterium breve*, *B. longum*, *B. infantis*) and lactobacilli (*Lactobacillus acidophilus*, *L. plantarum*, *L. paracasei*, *L. delbrueckii* subsp. *bulgaricus*), at very high concentration (450 billion bacteria).

The physical activity program was based on the individual difficulty of movement (assessed with a visual analogic scale) and possible pain. The motor activity program included a 1-h outdoor walk aerobic activity with 15–45 min of ergometer group gymnastics for 5 days/week.

In the morning after an overnight fast, basal metabolic rate was assessed with indirect computerized calorimetry (Vmax 29, Sensor Medics, Yorba Linda, CA, USA) and body composition by bioelectrical impedance analysis (BIA 101 Anniversary, Akern, Florence, Italy).

Anthropometric data (weight, height, body mass index—BMI, waist circumference) and fasting blood samples were collected on the admission to the hospital (T0), after 15 days of balanced hypocaloric Mediterranean diet (T1) and after two additional weeks of diet with probiotic mix supplementation (T2). Blood samples were used for the measurement of inflammatory markers (high-sensitivity C-reactive protein (hs-CRP), interleukin 6 (IL-6), tumor necrosis factor alpha (TNFα)), metabolic parameters (fasting glucose, insulin, HbA1c, lipids, AST, ALT, gamma-GT, uric acid), creatinine, adipokines (leptin, adiponectin), and oxidative stress markers (reduced glutathione (GSH), oxidized glutathione (GSSG), malondialdehyde (MDA)), as detailed below.

Dietary habits were collected by a trained dietitian through interviews using the 7-day diet history method to assess the frequency and servings of consumed foods.

Subjects were asked to collect a fecal sample 1–2 days before admission to the hospital (T0), and at T1 and T2, for gut microbiota analysis and calprotectin measurement. Fecal mass was classified using the Bristol stool scale (BSS), as developed and validated by Lewis and Heaton [25] and widely applied in both clinical practice and research. Samples were immediately frozen at −20 °C. For the gut microbiota analysis, samples were delivered to the Unit of Microbial Ecology of Health (Department of Pharmacy and Biotechnology, University of Bologna, Bologna, Italy) where they were stored at −80 °C until processing.

The Ethics Committee of Istituto Auxologico Italiano approved the study, and all subjects gave their written informed consent (ethical approval code: 2015_12_15_03).

### 2.2. Biochemical Analyses

Glucose, lipids, uric acid, creatinine, AST, ALT, gamma-GT and hs-CRP were measured using an automated analyzer (Roche Diagnostics, Mannheim, Germany). Leptin and adiponectin concentrations were quantified using a commercially available ELISA kit (Mediagnost, Reutlingen, Germany) with sensitivities of 0.2 and 0.27 ng/mL, respectively; overall, the inter- and intra-assay coefficients of variability (CVs) were <10%. Insulin was measured by a chemiluminescent assay (Roche Diagnostics). IL-6 and TNFα were measured by a commercially available ELISA method (Quantikine HS; R&D Systems Europe, Abingdon, UK), having sensitivities of 0.039 and 0.106 pg/mL, respectively. The intra- and inter-assay CVs were <10%.

GSH, GSSG and MDA were determined by high-performance liquid chromatography (HPLC) using kits from Chromsystems (Chromsystems Instruments & Chemicals, Gräfelfing, Germany). The sensitivities were 30 µmol/L for GSH and GSSG, and 0.01 µmol/L for MDA. The intra-and inter-assay CVs were <10% for GSH and GSSG, and <8% for MDA. Calprotectin protein was measured in fecal samples by Quantum Blue^®^ Calprotectin High Range Test (BÜHLMANN Laboratories AG, Schönenbuch, Switzerland). The normal range of calprotectin is 100–1800 µg/g.

### 2.3. Microbial DNA Extraction and 16S rRNA Gene-Based Illumina MiSeq Sequencing

Microbial DNA was extracted from feces using the repeated bead-beating plus column method [26], with only a few changes [27]. In short, about 250 mg of fecal sample was resuspended in 1 mL of lysis buffer (500 mM NaCl, 50 mM Tris-HCl pH 8, 50 mM EDTA, 4% SDS) and treated three times in a FastPrep instrument (MP Biomedicals, Irvine, CA, USA) at 5.5 movements/s for 1 min, in the presence of four 3-mm glass beads and 0.5 g of 0.1-mm zirconia beads (BioSpec Products, Bartlesville, OK, USA). After 15 min of incubation at 95 °C with pelleting stool particles at 13,000 rpm for 5 min, supernatants were added with 260 µL of 10 M ammonium acetate and incubated on ice for 5 min. After a further centrifugation step at 13,000 rpm for 10 min, one volume of isopropanol was added to each sample and incubated on ice for 30 min. Nucleic acid pellets were washed with 70% ethanol and then resuspended in 10 mM Tris-HCl, 1 mM EDTA pH 8.0 (TE) buffer. After treatment with 10 mg/mL DNase-free RNase at 37 °C for 15 min, samples were subjected to protein removal and column-based DNA purification following the manufacturer’s instructions (QIAamp DNA Stool Mini Kit; QIAGEN, Hilden, Germany).

The V3–V4 hypervariable region of the 16S rRNA gene was amplified using the 341F and 805R primers with added Illumina adapter overhang sequences, as previously reported [28]. Amplicons were purified with a magnetic bead-based clean-up system (Agencourt AMPure XP; Beckman Coulter, Brea, CA, USA). Indexed libraries were prepared by limited-cycle PCR using Nextera technology, further cleaned up as described above, and pooled at equimolar concentration. The final library was denatured with 0.2 N NaOH and diluted to 6 pM with a 20% PhiX control. Sequencing was performed on an Illumina MiSeq platform using a 2 × 300 bp paired-end protocol, according to the manufacturer’s instructions (Illumina, San Diego, CA, USA). Sequencing reads were deposited in MG-RAST (project ID, mgp91718).

### 2.4. Bioinformatics and Statistics

Comparisons of clinical/biochemical data at the three time points (T0, T1 and T2) were performed with one-way ANOVA for repeated measures. Pairwise comparisons of means were conducted by the Tukey method to take into account the type I error (alpha). Data were expressed as mean ± SD unless specified. A *p*-value < 0.05 was considered statistically significant. All analyses were performed using JMP (SAS v.12).

As for the gut microbiota, raw sequences were processed using a pipeline combining PANDAseq [29] and QIIME [30]. High-quality reads were binned into OTUs at 97% similarity using UCLUST [31]. Taxonomy was assigned using the RDP classifier against the Greengenes database (released in May 2013). All singleton OTUs and chimeras (identified using ChimeraSlayer [32]) were discarded. 16S rRNA gene sequencing data of EO patients were compared with publicly available data of 15 Italian non-obese elderly subjects (eight males and seven females, aged 72.5 ± 3.7 years) from a previous study (MG-RAST ID: 17761) [27] as controls (EC). According to the original publication [33], EC were non-institutionalized and living in their own household, showed good cognitive and physical health conditions, and a low-inflammation profile comparable to that of young adults. Genus-level community composition was generated for combined cohorts. Alpha diversity was computed using the inverse Simpson index. Beta diversity was estimated by computing Bray–Curtis distances, which were used as input for Principal Coordinate Analysis (PCoA). All statistical analysis was performed in R 3.3.2 using R studio 1.0.136. PCoA and Adonis tests (permutation test with pseudo-F ratios) were performed using the vegan package [34]. Correlation (Kendall tau) tests and non-parametric tests (Kruskal–Wallis test or Wilcoxon test, paired or unpaired as needed) were achieved using the stats package [35]. Only correlations with absolute Kendall’s tau ≥0.2 were considered. *P*-values were corrected for multiple comparisons using the Benjamini–Hochberg method when appropriate. A *p*-value ≤ 0.05 was considered statistically significant; a *p*-value between 0.05 and 0.1 was considered a tendency.

## 3. Results

### 3.1. Clinical and Biochemical Characteristics of Elderly Obese Women

Twenty elderly obese (EO) women were enrolled (aged 79.1 ± 3.3 years). Two (10%) EO patients had BMI between 30 and 34.9 kg/m^2^ (class I obesity hereinafter referred to as OB1 group; aged 82 ± 7 years), nine (45%) between 35 and 39.9 kg/m^2^ (class II obesity, OB2 group; aged 79.4 ± 2.7 years) and nine (45%) had BMI ≥ 40 kg/m^2^ (class III obesity, OB3 group; aged 78.1 ± 1.5 years). Seventy-five percent of patients had arthrosis, 80% were hypertensive, 30% had an obstructive sleep apnea and 25% had type 2 diabetes; 53% declared chronic constipation and 2% were active smokers. Table 1 summarizes the clinical and biochemical characteristics of the 20 EO patients.

Apart from the obvious differences in weight, BMI and waist circumference, it is worth noting that the OB2 and OB3 groups also differed by leptin, whose levels were significantly higher in the latter (*p* = 0.003, Wilcoxon test).

The home dietary habits showed an excessive consumption of fruit, pasta and bread, processed meat (ham) and cheese, a low intake of vegetables and legumes, and habitual consumption of wine (Appendix A).

### 3.2. Gut Microbiota Layout in Elderly Obese Women

The gut microbiota of EO women was profiled and compared with that of healthy non-obese elderly Italians, whose data are publicly available [27]. A total of 3,424,272 high-quality reads (mean ± SD, 57,071 ± 13,683) were obtained and analyzed.

No differences were observed in alpha diversity between EO and EC subjects, except for a tendency to decreased biodiversity in the first (*p* = 0.1, Wilcoxon test), which was significant in those with class II obesity (i.e., OB2) (*p* = 0.03) (Figure 1A). The Principal Coordinate Analysis (PCoA) of inter-individual variation, based on Bray–Curtis distances between the genus-level microbial profiles, nevertheless showed a significant separation between the study groups (*p* = 1 × 10^−5^, permutation test with pseudo-F ratios) (Figure 1B). In line with the available literature on the gut microbiota in obesity and metabolic disorders in general [36,37,38,39,40,41], EO patients showed increased relative abundance of *Coriobacteriaceae*, especially *Collinsella*, and *Streptococcus*, as well as reduced proportions of a number of SCFA producers belonging to the *Lachnospiraceae* and *Ruminococcaceae* families, including the understudied *Oscillospira* (*p* ≤ 0.05, Wilcoxon test). In addition, significantly lower amounts of *Parabacteroides*, a bacterial genus to which anti-obesogenic effects have recently been attributed [42], were found in EO vs. EC subjects (*p* ≤ 0.004) (Figure 2).

Consistently, in the EO cohort, we observed a negative correlation between the baseline relative abundance of *Lachnospiraceae* and the obesity-related anthropometric parameters BMI (tau = −0.301, *p* = 0.06, Kendall’s rank correlation test) and waist circumference (tau = −0.351, *p* = 0.03). In line with a recent population-based study on visceral fat accumulation [43], this was particularly evident for one of its members, i.e., *Blautia* (BMI, tau = −0.364, *p* = 0.03; waist circumference, tau = −0.33, *p* = 0.04), which was also found to be inversely related to HbA1c levels (tau = −0.324, *p* = 0.05). Inverse correlations were also found between *Faecalibacterium*, a butyrate producer from the *Ruminococcaceae* family with anti-inflammatory and immunomodulatory properties [44], already known to be less represented in obese, metabolic syndrome and type 2 diabetes patients [45], and weight (tau = −0.326, *p* = 0.05), BMI (tau = −0.354, *p* = 0.03), waist circumference (tau = −0.383, *p* = 0.02), fasting glucose (tau = −0.34, *p* = 0.04) and leptin (tau = −0.337, *p* = 0.04). Trends towards negative correlations were finally observed for *Methanobrevibacter* (tau = −0.284, *p* = 0.1) and *Akkermansia* (tau = −0.269, *p* = 0.1) against BMI (Appendix A). While some doubts remain on the association between methanogens and obesity, even if the current evidence is in favor of a positive correlation with a lean phenotype [46,47], *Akkermansia* is undoubtedly associated with metabolic benefits and its supplementation has recently been evaluated for the first time in humans, resulting in the improvement of several metabolic parameters in overweight/obese insulin-resistant volunteers [48]. On the other hand, confirming the available literature [39,49], the baseline relative abundance of *Coriobacteriaceae* was found to be directly related to waist circumference (tau = 0.277, *p* = 0.09).

### 3.3. Effect of Diet and Diet Supplemented with a Probiotic Mix in Elderly Obese Women

At the end of the rehabilitation program, EO patients experienced a weight loss of (4.0 ± 2.2)%. The greatest weight loss was recorded after the first 15 days of Mediterranean diet: by (2.7 ± 1.5)% in the whole group ((1.95 ± 1.5)% in class I, (4.2 ± 2.2)% in class II and (2.8 ± 1.3)% in class III obesity). Over the next 15 days of diet and probiotics, EO patients underwent a further weight loss by (1.2 ± 1.1)% ((1.4 ± 1.2)% in class I, (1.3 ± 1.2)% in class II and (1.4 ± 1.6)% in class III obesity).

In Appendix A, the changes in food consumption (weekly times) between home and rehabilitation are shown. Expectedly, the consumption of vegetables increased while that of pasta, bread, fruit, processed meat (ham), cheese and wine decreased.

There was a significant decrease in LDL cholesterol (82.5 ± 25.7 mg/dL at T1 and 88.6 ± 28.9 mg/dL at T2, *p* < 0.001 vs. baseline) and a trend towards fasting glucose decrease (100.9 ± 13.1 mg/dL at T1 and 103.3 ± 14.1 mg/dL at T2, *p* < 0.1 vs. baseline). With specific regard to oxidative stress markers, there was a progressive increase in GSH over time, with an increase in the GSH/GSSG ratio at the end of the intervention (*p* < 0.05 vs. baseline) (Figure 3A,B). On the other hand, the circulating levels of MDA, inflammatory markers and adipokines did not significantly change, except for a reduction in leptin levels (T0 vs. T1, *p* = 0.07). Fecal consistency improved, with an increase in types 3 and 4 (i.e., normal feces) and a decrease in types 5 and 6 (semi-formed/mushy stools) according to the Bristol stool scale classification (Figure 3C). In parallel, calprotectin levels decreased significantly after diet and probiotic supplementation (*p* = 0.05) (Figure 3D). Appendix A summarizes the clinical and biochemical characteristics for all patients after dietary intervention.

As for the gut microbiota, a significant increase in alpha diversity, comparable to that of EC subjects, was observed after 15 days of the rehabilitation program (i.e., after balanced hypocaloric Mediterranean diet) in EO patients with class II obesity (i.e., OB2) (*p* = 0.04, Wilcoxon test) (Figure 4A). PCoA analysis of Bray–Curtis distances between the genus-level microbial profiles showed no segregation among groups (*p* > 0.05, permutation test with pseudo-F ratios) but a significant shift towards decreasing PCo1 values was observed at T1 compared to the baseline (*p* = 0.03, Wilcoxon test) (Figure 4B), suggesting an impact, albeit limited, on the microbiota structure.

Interestingly, at the compositional level, some obesity-related dysbiotic features were already reversed after 15 days of intervention and others tended to be (Figure 5). In particular, the relative abundance of *Collinsella* was significantly reduced while that of *Parabacteroides* increased approximately 2-fold (*p* ≤ 0.04). It is worth noting that these changes remained significant (*Collinsella*, *p* = 0.04) or tended to be (*Parabacteroides*, *p* = 0.07) until the end of the intervention and were overall more pronounced in EO patients with class II obesity (i.e., OB2). Increasing trends at T1 were also observed for *Coprococcus* and *Oscillospira* (SCFA producers belonging to the *Lachnospiraceae* and *Ruminococcaceae* families, respectively) (*p* = 0.08), whose baseline levels were lower compared to EC subjects. In particular, the significance for *Coprococcus* was reached only at T2 (*p* = 0.003). In addition, other bacteria typically related to leanness [47,50,51,52] showed various but significant increases in percentage, i.e., *Bacteroides* (T0 vs. T1, *p* = 0.03), *Christensenellaceae* (T0 vs. T1, *p* = 0.03; T0 vs. T2, *p* = 0.01), *Methanobrevibacter* (T0 vs. T1, *p* = 0.04) and *Akkermansia* (T0 vs. T2, *p* = 0.03). Again, the changes in *Bacteroides* and *Akkermansia* reached significance only in the OB2 group (Figure 5). On the other hand, in OB2 women, we observed a significant decrease for *Enterococcus* (T0 vs. T1, *p* = 0.02), typically enriched in the gut microbiota of obese subjects, associated with Western diets and supposed to mediate their detrimental effects on colonic health [53,54]. Though not significant, the proportions of *Faecalibacterium* tended to increase at the end of the intervention in EO patients with class III obesity, OB3 (*p* = 0.07). The relative abundance of most of these bacterial taxa showed consistent correlations with weight, BMI and calprotectin levels in the whole cohort (Appendix A).

Supporting patient compliance, after diet and probiotic supplementation, there was an increase in the relative abundances of bifidobacteria, lactobacilli and streptococci, i.e., the same bacterial genera included in the probiotic mixture (Appendix A).

## 4. Discussion

In this short-term dietary intervention study in elderly obese women (EO), we showed that two weeks of balanced Mediterranean diet with mild caloric deficit, as a part of a metabolic-nutritional-psychological rehabilitation program, improved the patients’ metabolic picture, and that such an improvement was accompanied by the recovery of a balanced health-promoting configuration of the gut microbiota. The Mediterranean diet is indeed considered the optimal strategy to prevent microbiota dysbiosis and protect intestinal permeability [55]. In line with the literature available on gut microbiota and metabolic disorders [36,37,38,39,40,41], the intestinal microbiota of EO subjects, as compared to non-obese elders (EC) living in the same territory (i.e., Italy), showed several dysbiotic features, including: (i) a tendency to reduced biodiversity (generally recognized as a hallmark of a healthy gut); (ii) decreased relative abundance of health-promoting, SCFA producers (mainly belonging to the *Lachnospiraceae* and *Ruminococcaceae* families, i.e., *Lachnospira*, *Blautia*, *Coprococcus*, *Roseburia*, *Ruminococcus* and *Oscillospira*) as well as *Parabacteroides*; and (iii) increased proportions of subdominant taxa, such as *Collinsella* and *Streptococcus*. While SCFAs are well known to have a key, multifactorial role in the host physiology, being fundamental for the maintenance of metabolic and immunological homeostasis [56], *Collinsella* and *Streptococcus* have been hypothesized to be involved in a variety of inflammation-based disorders [39,57,58,59,60]. In particular, increased levels of *Collinsella* have so far been found in overweight and obese individuals (including pregnant women) as well as in patients suffering from type 2 diabetes and symptomatic atherosclerosis, and are generally associated, albeit with some discordance, with a number of metabolic parameters, including insulin, triglycerides and LDL cholesterol [37,39,59]. The underlying molecular mechanisms are still unknown but are likely to involve alteration of intestinal cholesterol absorption, reduction of glycogenesis in the liver and increase in triglyceride synthesis, as well as reduction of the expression of tight junction proteins, possibly leading to gut leakage and metabolic endotoxemia [37,61]. Although more direct evidence of *Collinsella* being pro-inflammatory, interfering with our metabolism and therefore being harmful to human health is needed, with the wave of enthusiasm in recent studies, *Collinsella* has been proposed as a target in future microbiome-based interventions for metabolic disorders [36]. A completely different role has instead been established for *Oscillospira* and *Parabacteroides*, with the first probably representing a heritable taxon capable of promoting leanness [40,47], and the second recently suggested as anti-obesogenic probiotics, capable of increasing adipose tissue thermogenesis, enhancing intestinal integrity and reducing levels of inflammation and insulin resistance [42]. It is also worth mentioning that *Blautia* has recently been identified as the only gut microbe significantly and inversely associated with visceral fat accumulation regardless of sex, in a Japanese population-based study [43].

As mentioned above, after two weeks of balanced hypocaloric Mediterranean diet, the dysbiotic microbiota signatures of obesity were largely reversed, and patients experienced modest weight loss, and reductions of LDL cholesterol and fasting blood glucose. Several weight-loss trials demonstrated that weight loss in elderly people may be dangerous due to the losses of lean body mass and bone mineral density, in addition to fat mass [62]. However, in EO subjects, it is essential to apply a weight loss strategy because the excessive adiposity contributes to frailty by reducing the ability to perform physical activities and increasing chronic inflammation and related comorbidities [63]. Furthermore, there is strong clinical evidence that weight reduction improves physical function and metabolic and cardiovascular parameters in EO patients [62,63]. With specific regard to the gut microbiota, the balanced hypocaloric diet was found to counteract the rise in potential pathobionts, namely *Collinsella*, as well as the decrease in SCFA producers, mainly *Coprococcus* and *Oscillospira*, and in *Parabacteroides*, with a significantly increased biodiversity in EO patients with class II obesity. Further confirming the available literature on the association between intestinal microbiota components and obesity, two weeks of intervention also led to increases in other bacterial taxa typically related to leanness, i.e., *Bacteroides* [50], *Christensenellaceae* [47] and *Methanobrevibacter* [50], with the latter two showing consistent inverse correlations with obesity-related anthropometric/biochemical parameters (i.e., BMI and calprotectin levels). In particular, previous studies support the presence of methanogens as a marker for a low BMI in humans but strongly suggest that members of *Christensenellaceae*, being recognized as the most heritable taxa of the human gut microbiome, are required to promote a lean host phenotype [47].

Adding probiotics to the diet over the next two weeks reduced constipation and calprotectin levels, and enhanced diet-induced oxidative stress improvement. From the microbiota standpoint, apart from the consolidation of healthy-like relative abundances of some bacteria as mentioned above (i.e., *Collinsella*, *Coprococcus* and *Christensenellaceae*) and the obvious increases in the same bacterial genera represented in the probiotic mixture (*Bifidobacterium*, *Lactobacillus* and *Streptococcus*), we also found a significant increase in *Akkermansia*, a mucin-degrading microbe identified as a promising next-generation candidate for the development of novel food or pharma supplements for metabolic disorders [64]. The potential of *Akkermansia* to target specific metabolic health issues has been demonstrated in a very recent proof-of-concept exploratory study, where Depommier et al. [48] showed that daily oral administration of *Akkermansia muciniphila*, either live or pasteurized, for three months to overweight/obese insulin-resistant volunteers, was safe and well tolerated, and led to the improvement of multiple metabolic parameters. Despite the small sample size, it should be pointed out that most of these beneficial changes in the microbiota structure reached significance in patients with class II obesity, probably indicating that longer times or different conditions are needed in cases of extreme obesity.

## 5. Conclusions

In conclusion, this study demonstrated that two weeks of dietary intervention, as part of a metabolic-nutritional-psychological rehabilitation program, improved the metabolic picture and largely reversed the gut microbiota dysbiosis found in elderly obese women towards a profile typically associated with metabolic health. Adding probiotics to the diet further reduced oxidative stress and led to increased relative abundance of *Akkermansia*, a mucin degrader with well-recognized beneficial effects on host metabolism. While direct evidence on the relevance of specific microbiota components as targets for the prevention or treatment of metabolic disorders is mandatory, our findings strongly suggest that correct dietetic approaches, even in the short term, may be effective in adjusting unbalanced gut microbiota profiles towards more favorable configurations, instrumental to maintaining health. Generally, in old age, there is a lesser tendency to correct cardio-metabolic risk factors with pharmacological therapy because of possible adverse effects. It is therefore important to know that the combined use of balanced Mediterranean diet and probiotics has the potential to induce substantial beneficial effects on the gut microbiota and risk profile even in individuals with high cardio-metabolic risk, such as the elderly obese patient. Qualitative changes in dietary habits with a modest caloric deficit should not be difficult to maintain over time and possibly represent the appropriate treatment approach for elderly obese subjects. Future studies are required to validate these findings in larger cohorts, also including male elderly obese patients, to explore the possible causal role of these positive microbiota changes (or if they merely represent a biomarker indicating an improved metabolic state) and their stability over time, and to provide mechanistic insights on the interactions between the gut microbiota and our physiology.

## Figures and Tables

**Figure 1 nutrients-11-03011-f001:**
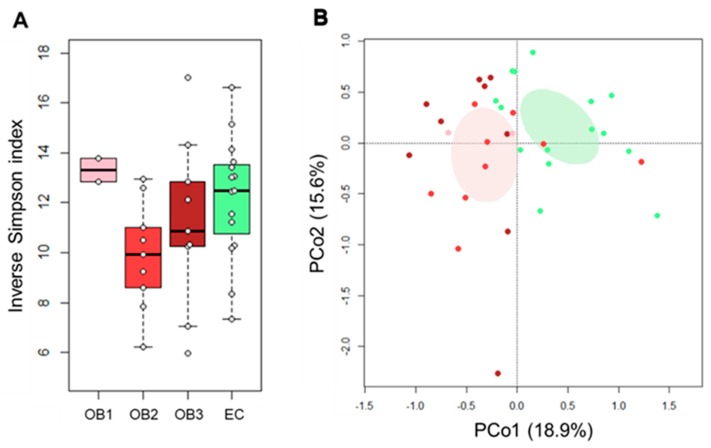
The gut microbiota of obese elderly women segregates from that of healthy elders. (**A**) Box plots showing the distribution of alpha diversity values, according to the inverse Simpson index, in elderly subjects with class I (OB1), class II (OB2) and class III (OB3) obesity, as compared to healthy non-obese controls (EC). Samples are identified with gray dots within boxes. A significantly reduced biodiversity was observed in OB2 patients compared to EC (*p* = 0.03, Wilcoxon test). (**B**) PCoA plot of inter-sample diversity, based on Bray–Curtis distances between the genus-level microbial profiles. A significant separation between obese elderly patients and EC was found (*p* = 1 × 10^−5^, permutation test with pseudo-F ratios). Samples are identified with colored dots as in A. Ellipses include 99% confidence area based on the standard error of the weighted average of sample coordinates (lilac, elderly obese patients; green, EC).

**Figure 2 nutrients-11-03011-f002:**
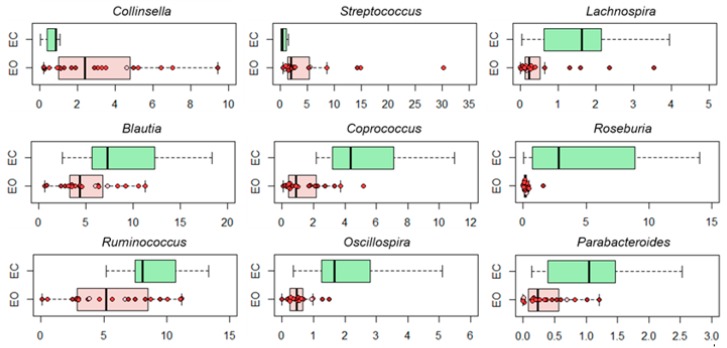
Genus-level compositional differences in the gut microbiota of obese vs. healthy elders. Box plots showing the distribution of the relative abundance values of discriminant genera between obese (EO) and non-obese healthy elders (EC). Samples from EO patients with class I (OB1, pink), class II (OB2, red) and class III (OB3, brown) obesity are identified with colored dots within boxes. *p* ≤ 0.05, Wilcoxon test.

**Figure 3 nutrients-11-03011-f003:**
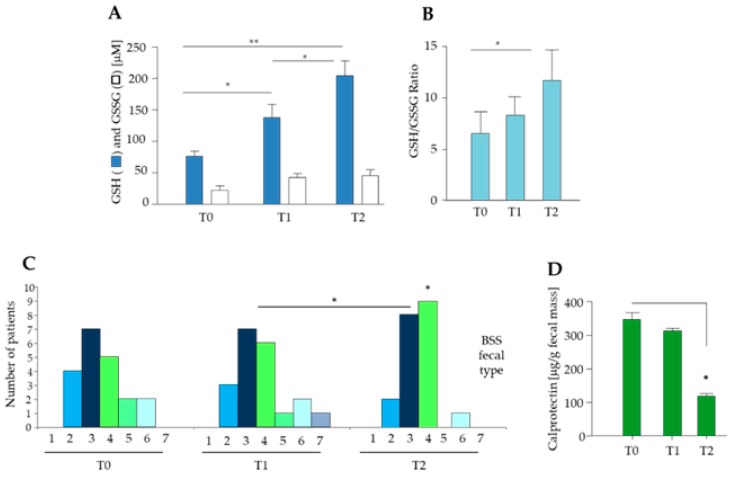
Diet and probiotic mix supplementation improve oxidative stress and stool consistency, and reduce fecal calprotectin levels in elderly obese women. (**A**) Circulating levels of reduced (GSH) and oxidized (GSSG) glutathione, (**B**) GSH/GSSG ratio, (**C**) stool consistency by Bristol stool scale (BSS) ranging from type 1 to type 7, where types 1–2 indicate constipation, types 3–4 ideal stools and types 5–7 diarrhea and urgency, and (**D**) fecal calprotectin concentrations in elderly obese patients before intervention (T0), after two weeks of hypocaloric balanced Mediterranean diet (T1) and after two additional weeks of diet with probiotic mixture supplementation (T2). Data are expressed as mean ± SD. * *p* < 0.05; ** *p* < 0.0001, ANOVA.

**Figure 4 nutrients-11-03011-f004:**
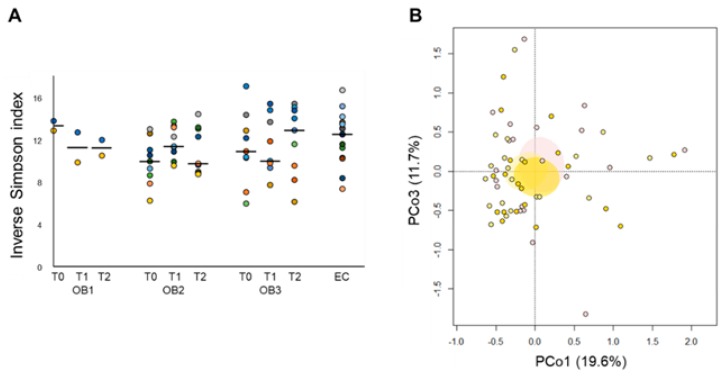
Diet and probiotic mix supplementation impact on the gut microbiota diversity of elderly obese women. (**A**) Alpha diversity values, according to the inverse Simpson index, in elders with class I (OB1), class II (OB2) and class III (OB3) obesity, before intervention (T0), after two weeks of hypocaloric balanced Mediterranean diet (T1) and after two additional weeks of diet with probiotic mixture supplementation (T2). On the right, as a control, the biodiversity of the gut microbiota of healthy non-obese elders (EC). Within each group (OB1 to OB3 and EC), samples are colored by subject. The black line within each series indicates the median value. A significantly increased biodiversity was observed in OB2 patients at T1 compared to the baseline (*p* = 0.04; Wilcoxon test). (**B**) PCoA plot based on Bray–Curtis distances between the genus-level microbiota profiles of EO patients at the baseline (pink), T1 (light yellow) and T2 (yellow). Ellipses include 99% confidence area based on the standard error of the weighted average of sample coordinates. No significant segregation was found (*p* > 0.05, permutation test with pseudo-F ratios).

**Figure 5 nutrients-11-03011-f005:**
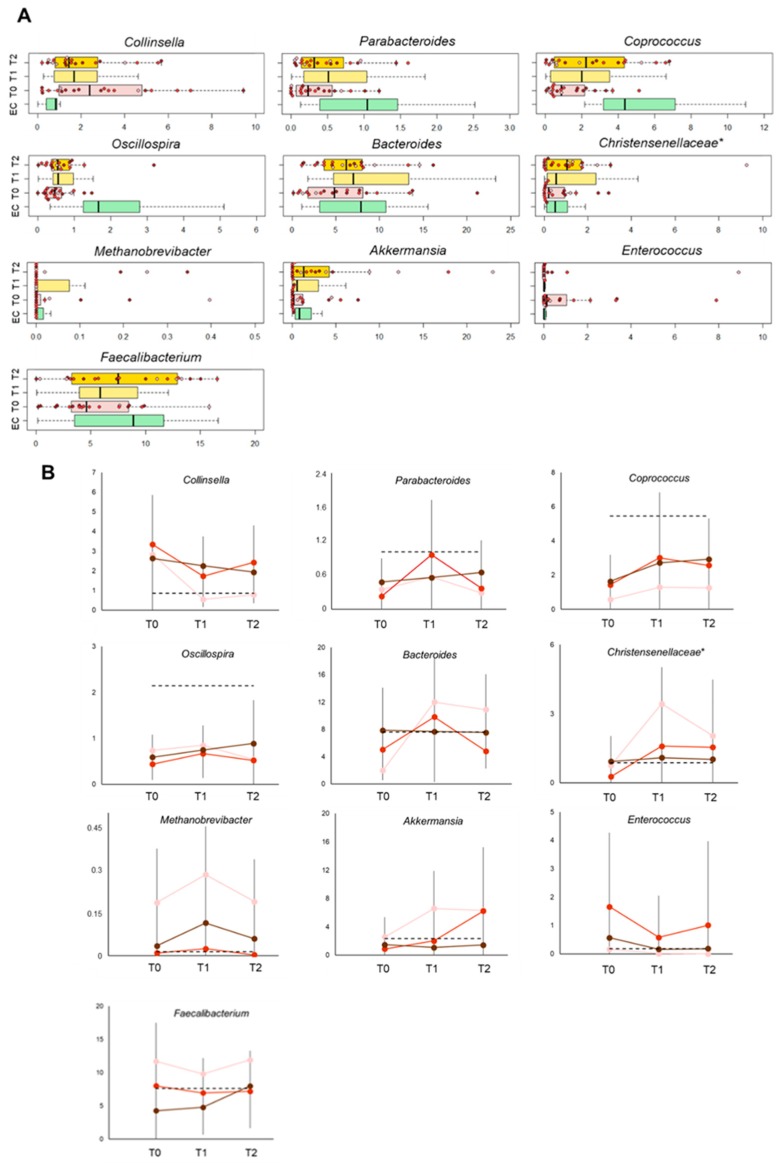
Diet and probiotic mix supplementation counteract the obesity-related dysbiotic features of the gut microbiota in elderly obese women. (**A**) Box plots showing the distribution of the relative abundance values of significantly altered bacterial genera over the course of the intervention (*p* ≤ 0.05, Wilcoxon test). For *Oscillospira* and *Faecalibacterium*, only trends were observed (*p* ≤ 0.08). T0, before intervention; T1, after two weeks of hypocaloric balanced Mediterranean diet; T2, after two additional weeks of diet with probiotic mixture supplementation. Samples are identified with colored dots within boxes, based on the obesity grade as assessed at T0 and T2 (pink, class I obesity—OB1; red, class II obesity—OB2; brown, class III obesity—OB3). EC, healthy non-obese elderly. *, unclassified OTU reported at higher taxonomic level. (**B**) Relative abundance dynamics for the genera of panel A. Changes in mean values (±SD) are shown by obesity grade (same color code as in **A**). The dotted line indicates the mean value in the control population (i.e., EC).

**Table 1 nutrients-11-03011-t001:** Clinical and biochemical characteristics of the 20 elderly obese women enrolled (All) and the relative subdivision by body mass index (BMI) (kg/m^2^) ranges (30 ≤ OB1 < 35; 35 ≤ OB2 < 40; ≥40, OB3).

Parameter	All (*N* = 20)	OB1 (*N* = 2)	OB2 (*N* = 9)	OB3 (*N* = 9)
Age (years)	79.1 ± 3.3	82 ± 7	79.4 ± 2.7	78.1 ± 1.5
Arthrosis (*N*)	15	1	8	6
Hypertension (*N*)	16	1	8	7
Obstructive sleep apnea (*N*)	6	0	3	3
Type 2 diabetes (*N*)	5	0	2	3
Chronic constipation (*N*)	12	2	5	5
Weight (kg)	93.8 ± 14.5	78.6 ± 6.6	87.1 ± 5.1	104.9 ± 13.6 ^a,b^
Height (cm)	152.4 ± 6.1	147.7 ± 1.7	152.6 ± 5.1	153.1 ± 7.1
BMI (kg/m^2^)	40.3 ± 4.9	33.8 ± 0.05	37.3 ± 1.4 ^c^	44.7 ± 3.7 ^a,b^
Waist circumference (cm)	116.6 ± 10.9	102.0 ± 7	110.1 ± 6.1 ^c^	126.3 ± 5.6 ^a,b^
Fat mass (%)	50.1 ± 3.4	48.5 ± 3.1	49.6 ± 3.8	50.8 ± 2.6
Fat free mass (%)	49.9 ± 3.4	51.5 ± 3.1	50.4 ± 3.9	49.1 ± 2.6
Fasting glucose (mg/dL)	107.5 ± 20.3	108.5 ± 2.5	99 ± 14.3	115.8 ± 23.8
Insulin (μU/mL)	13.8 ± 8.5	28.1 ± 12.9	10.9 ± 4.9	13.58 ± 6.72
HbA1c (mmol/mol)	5.8 ± 0.5	5.9 ± 0.4	5.5 ± 0.3	6.1 ± 0.63
Total cholesterol (mg/dL)	190.9 ± 22.9	207.5 ± 8.5	195 ± 24.1	182.12 ± 21.8
LDL cholesterol (mg/dL)	121.5 ± 27.3	144 ± 3	126.4 ± 26.1	111.5 ± 27.1
HDL cholesterol (mg/dL)	58.9 ± 16.3	51 ± 10	57.1 ± 16.1	62.3 ± 16.8
Triglycerides (mg/dL)	124.5 ± 45.5	143 ± 2	138.1 ± 57.6	104.5 ± 22.1
hs-CRP (mg/dL)	0.7 ± 0.7	0.7 ± 0.5	0.8 ± 0.9	0.53 ± 0.4
AST (U/L)	15.9 ± 2.6	17.5 ± 0.5	15.8 ± 2.1	15.5 ± 3.1
ALT (U/L)	14.7 ± 3.0	17 ± 0.1	15.0 ± 2.2	13.7 ± 3.6
Gamma-GT (U/L)	20.4 ± 9.0	32 ± 15	20.1 ± 6.4	17.7 ± 7.1
Uric acid (mg/dL)	5.7 ± 1.2	5.1 ± 0.1	6.2 ± 0.9	5.3 ± 1.2
Creatinine (mg/dL)	0.9 ± 0.2	0.9 ± 0.1	0.75 ± 0.1	0.96 ± 1.2
IL-6 (pg/mL)	3.9 ± 2.1	2.4 ± 3.6	4.7 ± 2.9	3.1 ± 0.9
TNFα (pg/mL)	1.15 ± 0.5	1.2 ± 2.0	0.87 ± 0.24 ^c^	1.18 ± 0.5 ^a^
Leptin (ng/mL)	77.3 ± 39.8	35.4 ± 31.8	54.4 ± 18.3	105.9 ± 40.5 ^a,b^
Adiponectin (µg/mL)	14.5 ± 8.1	13.9 ± 4.8	17.7 ± 8.9	12.2 ± 6.2
Reduced glutathione, GSH (µmol/L)	78.0 ± 38.4	54.8 ± 0.0	74.7 ± 45.4	84.6 ± 29.7
Oxidized glutathione, GSSG (µmol/L)	22.2 ± 12.9	21.5 ± 0.0	21.7 ± 12.4	24.3 ± 14.2
Malondialdehyde, MDA (µmol/L)	7.3 ± 1.7	5.85 ± 0.85	7.5 ± 2.2	7.3 ± 1.13
Calprotectin (µg/g)	376.8 ± 420.9	182.5 ± 132.5	450.5 ± 530.8	346.3 ± 305.3

*N*, number. Data are expressed as mean ± SD. *p* < 0.05; ^a^, OB3 vs. OB2; ^b^, OB3 vs. OB1; ^c^, OB2 vs. OB1.

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
