# Peer review of "Effect of Short-Term Dietary Intervention and Probiotic Mix Supplementation on the Gut Microbiota of Elderly Obese Women"

_nutrients, 2019, doi:10.3390/nu11123011_

Round 1

Reviewer 1 Report

General comment:

Effect of short-term dietary intervention and probiotic mix supplementation on the gut microbiota of elderly obese subjects by Cancello et al described the effect of VSL#3 probiotic mixture on elderly obese subjects after 2 weeks of treatment. While the experimental approach looks solid at first glance, I'm questioning the low number of subjects and most of it the gender bias (20 females and 3 males) and the soundness of the statistical approach.

Specific comments:

-Line172: sequencing reads depository number is missing which makes it impossible to review.

-Line186: the use of "Simpson index" is not correct because the authors are interpreting their results like they used "inverse Simpson index". (Line 219)

-Line189: reference for vegan package is missing.

-Line 191: reference for stats package is missing.

-Line197: there is an obvious gender bias. I would suggest to remove the 3 males subjects and only keep the 20 females. The titel should be adapted to make it more specific to female.

-Line219: decreased Simpson index (figure 1A) means increased diversity. Indeed low Simpson index refers to high diversity. The authors interpretation is related to the use of Inverse Simpson. Please clarify the diversity index used.

-Line244: figure 1A refers to inverse Simpson index, not Simpson index. Please correct it.

-Line297: figure 4A refers to inverse Simpson index, not Simpson index. Please correct it.

-Line309: figure 5 about obesity-related dysbiotic feature should include the "EC" population as for figure 1 and figure 4.

-Line312-314: remaining significant until the end of the intervention is only true for OB2. Any reasons for that? Please rephrase accordingly and double-check everywhere that you statement are backed-up by your (graphical) data.

-Line320: Faecalibacterium data are not presented in supplementary Figure S4. Please clarify.

-Line337: please rephrase by removing "(though not always significant)" and "lactobacilli"

Author Response

REVIEWER1

General comment:

Effect of short-term dietary intervention and probiotic mix supplementation on the gut microbiota of elderly obese subjects by Cancello et al described the effect of VSL#3 probiotic mixture on elderly obese subjects after 2 weeks of treatment. While the experimental approach looks solid at first glance, I'm questioning the low number of subjects and most of it the gender bias (20 females and 3 males) and the soundness of the statistical approach.

We would like to thank the Reviewer for her/his remarks we have tried to address in the revised version of our manuscript.

Specific comments:

-Line172: sequencing reads depository number is missing which makes it impossible to review.

We apologize for not having provided the reviewer token before.

We deposited the sequencing reads in MG-RAST and included the project ID (mgp91718) in the main text (L173).

-Line186: the use of "Simpson index" is not correct because the authors are interpreting their results like they used "inverse Simpson index". (Line 219)

We apologize for not having used the correct terminology.

In the previous version of our manuscript, we did not use the Simpson’s original index (D) but its complement, 1-D, whose value increases as diversity increases.

In any case, given that the inverse Simpson index is more commonly used, we re-analyzed the data and showed the alpha diversity values as inverse of Simpson’s index. Please, see the revised text (L192,263,324) and the new figures 1A and 4A.

-Line189: reference for vegan package is missing.

We apologize for the oversight. The corresponding reference has been added to the revised version of the manuscript (L195, ref. 34).

-Line 191: reference for stats package is missing.

As above, the related reference has been added (L197, ref. 35).

-Line197: there is an obvious gender bias. I would suggest to remove the 3 males subjects and only keep the 20 females. The titel should be adapted to make it more specific to female.

We totally agree that gender may represent a potential bias: woman's reproductive status, menstrual cycle and contraceptive history have become significant in studying health, disease and pharmacology. However, in our study protocol, this aspect most probably represents a minor confounding factor since we were working with post-menopausal elderly obese women.

In any case, in the light of the possible existence of gender-specific outcomes in nutritional rehabilitation, we decided to exclude, as requested, the only three male patients enrolled. All the data has been re-analyzed, and title and text (including tables and figures) have been changed accordingly.

-Line219: decreased Simpson index (figure 1A) means increased diversity. Indeed low Simpson index refers to high diversity. The authors interpretation is related to the use of Inverse Simpson. Please clarify the diversity index used.

As discussed above, in the previous version of our manuscript, we actually used the complement of the Simpson’s original index (i.e. 1-D), whose values are directly correlated to biodiversity. We apologize again for the inaccuracy. In any case, in the revised version of our paper, we decided to use the inverse Simpson’s index since it is much more used in human gut microbiota studies. Please, see the revised text (L192,263,324) and the new figures 1A and 4A.

-Line244: figure 1A refers to inverse Simpson index, not Simpson index. Please correct it.

Please, see the answer above.

-Line297: figure 4A refers to inverse Simpson index, not Simpson index. Please correct it.

Please, see the answer above.

-Line309: figure 5 about obesity-related dysbiotic feature should include the "EC" population as for figure 1 and figure 4.

Following the Reviewer’s suggestion, we have included the control population (i.e. EC) in Figure 5A.

-Line312-314: remaining significant until the end of the intervention is only true for OB2. Any reasons for that? Please rephrase accordingly and double-check everywhere that you statement are backed-up by your (graphical) data.

We apologize for the confusion.

As correctly stated by the Reviewer, the changes that were and remained significant refer to the OB2 group. Not considering the group OB1, for obvious reasons related to the size of the sample, this data probably suggests that 30 days of metabolic-nutritional-psychological rehabilitation program, as described in this work, are not sufficient or totally adequate in case of extreme obesity (please, see L442-444).

Following the Reviewer’s suggestion, we have rephrased where appropriate and double checked that, in the revised version of our manuscript, all sentences are consistent with the graphic data.

-Line320: Faecalibacterium data are not presented in supplementary Figure S4. Please clarify.

The brief comment on the variation of the relative abundance of Faecalibacterium was introduced (despite the non-significance, p=0.07) in light of the well-known anti-inflammatory and immunomodulatory properties of this next-generation probiotic candidate. We believe that this increase is worthy of being cited, especially with a view to a longer treatment, as our data on OB3 seem to suggest. Accordingly, in the revised version of our manuscript, we included Faecalibacterium in Figure 5.

As for correlations, based on the new dataset (i.e. 20 females), significant inverse correlations were found between the baseline relative abundance of Faecalibacterium and weight, BMI, waist circumference, fasting glucose and leptin (please, see the new Supplementary Figure S2). The correlations with weight and BMI were also significant (p=0.05) in the whole cohort but with absolute Kendall’s tau <0.2 (-0.174 and -0.171, respectively) and as such, we chose not to include the corresponding scatter plots in Supplementary Figure S3.

-Line337: please rephrase by removing "(though not always significant)" and "lactobacilli"

We thank the Reviewer for these suggestions. We have removed the term “though not always significant” but kept “lactobacilli” based on the new analysis (L373 and Supplementary Figure S4).

Reviewer 2 Report

The manuscript entitled, “Effect of short-term dietary intervention and 2 probiotic mix supplementation on the gut microbiota 3 of elderly obese subjects” is a novel study demonstrating that the hypocaloric diet with probiotic mix is enough to improve metabolic signs and reverse the gut microbiota dysbiosis in elderly obese subjects.  Although the manuscript has nicely been written and results are clearly been presented, I have some minor concerns:

Page 5; line 201: Table I is showing N=9 in OB3 category while text indicates 11 females. Page 5; line 202: “had an apnea and 22%% had type 2…..”. Please delete extra “%”. Figure 5: labeling and color dots in Fig 5A are not visible. Please make it legible. Clinical and biological characteristic of all the subjects after dietary intervention at T1 and T2 time points should be presented in separate Table.

Author Response

REVIEWER 2

The manuscript entitled, “Effect of short-term dietary intervention and 2 probiotic mix supplementation on the gut microbiota 3 of elderly obese subjects” is a novel study demonstrating that the hypocaloric diet with probiotic mix is enough to improve metabolic signs and reverse the gut microbiota dysbiosis in elderly obese subjects. Although the manuscript has nicely been written and results are clearly been presented, I have some minor concerns:

Page 5; line 201: Table I is showing N=9 in OB3 category while text indicates 11 females.

The Reviewer is right and we apologize for the oversight.

Following the Reviewer 1’s comments in relation to the gender bias, in the revised version of our manuscript, we removed the three males and kept only the 20 females. The entire text along with Table 1 and figures have been modified accordingly. The current dataset is composed as follows: 2 OB1, 9 OB2 and 9 OB3.

Page 5; line 202: “had an apnea and 22%% had type 2…..”. Please delete extra “%”.

Thanks for checking. The typo has been corrected.

Figure 5: labeling and color dots in Fig 5A are not visible. Please make it legible.

The figure has been modified following the Reviewer’s suggestions.

Clinical and biological characteristic of all the subjects after dietary intervention at T1 and T2 time points should be presented in separate Table.

As requested, we added a new table as Supplementary Table S1 including the available clinical and biological data for all patients after dietary intervention (i.e. at T1 and T2).

Round 2

Reviewer 1 Report

Dear Authors,

thank you for revising your manuscript and for taking into account my comments.

Author Response

Dear Reviewer, 

thanks to you for the valuable and helpful comments.